# Potential Epigenetic Effects of Human Milk on Infants’ Neurodevelopment

**DOI:** 10.3390/nu15163614

**Published:** 2023-08-17

**Authors:** Giannoula Gialeli, Ourania Panagopoulou, Georgios Liosis, Tania Siahanidou

**Affiliations:** 1First Department of Pediatrics, Medical School, National & Kapodistrian University of Athens, 11527 Athens, Greece; giannoula.g@gmail.com (G.G.); niakinder@yahoo.gr (O.P.); 2Neonatal Intensive Care Unit, “Elena Venizelou” General and Maternal Hospital, 11521 Athens, Greece; gtliosis@yahoo.gr

**Keywords:** epigenetics, human milk, neurodevelopment, miRNAs, long non-coding RNAs, stem cells, microbiome

## Abstract

The advantages of human milk feeding, especially in preterm babies, are well recognized. Infants’ feeding with breast milk lowers the likelihood of developing a diverse range of non-communicable diseases later in life and it is also associated with improved neurodevelopmental outcomes. Although the precise mechanisms through which human milk feeding is linked with infants’ neurodevelopment are still unknown, potential epigenetic effects of breast milk through its bioactive components, including non-coding RNAs, stem cells and microbiome, could at least partly explain this association. Micro- and long-non-coding RNAs, enclosed in milk exosomes, as well as breast milk stem cells, survive digestion, reach the circulation and can cross the blood–brain barrier. Certain non-coding RNAs potentially regulate genes implicated in brain development and function, whereas nestin-positive stem cells can possibly differentiate into neural cells or/and act as epigenetic regulators in the brain. Furthermore, breast milk microbiota contributes to the establishment of infant’s gut microbiome, which is implicated in brain development via epigenetic modifications and key molecules’ regulation. This narrative review provides an updated analysis of the relationship between breast milk feeding and infants’ neurodevelopment via epigenetics, pointing out how breast milk’s bioactive components could have an impact on the neurodevelopment of both full-term and preterm babies.

## 1. Introduction

Almost fifty years ago, the international scientific community believed that a person’s health and the expression of non-communicable diseases was solely a matter of that individual’s gene pool, which was not influenced by external factors. However, primitive genetic patterns could not explain the explosive increase in cancer and other non-communicable metabolic disorders [1]. Barker’s hypothesis provided a revolutionary answer to this issue. Based on the observation that coronary heart disease, obesity and type 2 diabetes had a higher incidence in the poorest areas of England, Professor Barker was able to link low birth weight and poor prenatal conditions to adult disease [2]. The fetal origins of adult disease (FOAD) hypothesis of Professor Barker holds that the embryo’s genome exhibits developmental plasticity [3]. Stressors, such as malnutrition, may remodel the embryos genome in order to prepare it for adverse extrauterine conditions, thus allowing a single genotype to produce multiple phenotypes depending on intrauterine conditions [4]. Over the following years, the FOAD was extended to “the Developmental Origins of Health and Disease” (DOHaD) hypothesis, which suggests that environmental exposures during early life, in both prenatal and postnatal period, can permanently influence health and the vulnerability to disease in later life by “programming” the phenotype without altering the genotype [5,6,7,8]. This programming process involves heritable changes in gene expression, which are mediated through epigenetic modifications such as DNA methylation, histone modification, and the activation or silencing of genes associated with non-coding RNAs [6,9,10]. These epigenetic mechanisms are suspected to play a crucial role in developmental programming [11]. Maternal stressors such as obesity or malnutrition, smoking and diabetes, among others, are known triggers for epigenetic modifications in the offspring [6,12].

The majority of human development occurs in the first 1000 days starting from conception. This time period of perinatal programming is considered critical in determining further development and health [13,14]. Postnatally, human breast milk is known to reduce the probability of expression of a wide variety of non-communicable diseases [15,16]. Breast milk may modify the epigenetic mechanisms of the infants and influence their health intergenerationally [17,18]. It is hypothesized that breast milk promotes epigenetic modifications via its bioactive components, including growth factors, microbiota, stem cells, micro-RNAs (miRNAs) and long-non-coding-RNAs [16,17,19,20]. Several studies have also shown that breast milk feeding, especially with mother’s own milk, is associated with improved neurodevelopmental outcomes in both full-term and preterm infants [21,22,23], whereas longer duration of exclusive breastfeeding has been linked to higher intelligence quotients [24] and improved cognitive development [25,26]. A positive impact of breast milk feeding on structural brain development in preterm babies has been demonstrated using brain magnetic resonance imaging (MRI) [27].

The underlying mechanisms that explain the connections between the consumption of breast milk—particularly the mother’s own milk—and the subsequent neurodevelopmental outcomes, especially in the vulnerable population of very-low-birth-weight (VLBW, <1500 g) infants, have not yet been clarified. The potential epigenetic effects of human milk could mediate the associations between breast milk feeding and brain development/neurodevelopment. Interestingly, Xu et al. have recently demonstrated that the percentage of the mother’s own milk intake during the hospital stay of VLBW infants was linked to changes in DNA methylation (DNAm) patterns of genes related to neurodevelopment at 5.5 years of age. Certain DNAm variations were associated with differences in brain structure and intelligence quotient (IQ) [28].

In this article, we discuss the potential epigenetic role of miRNAs and long non-coding RNAs, stem cells and microbiome of human milk on infants’ neurodevelopment.

## 2. Methods

For this narrative review, a literature search was conducted using the databases PubMed, Medline, ScienceDirect and Google Scholar (last accessed on 4 August 2023). Specific keywords such as breast milk, epigenetics, non-coding RNAs, miRNAs, long non-coding RNAs, stem cells, microbiome, brain development, neurodevelopment, infants, and preterm birth or prematurity were used. The inclusion criteria were as follows: all types of articles, articles published in PubMed, and studies using both humans and animals. Articles not written in the English language, or for which full text was not available, or were grey literature were excluded. From the articles retrieved in the first round of search, additional articles were identified via a manual search among the cited references (Table 1).

## 3. MiRNAs

A number of recent publications have demonstrated that human milk contains components recently described as extracellular vesicles (EVs) [29]. Extracellular vesicles is a term for all phospholipid bilayer-enclosed particles that are released by cells into their environment and include exosomes and microvesicles [29]. Exosomes carry bioactive substances like proteins, DNA, messenger RNA (mRNA) and miRNAs [29,30]. Breast milk exosomes, being resistant to digestion [31], are able to transport their cargo and miRNAs to peripheral tissues via the systemic circulation and facilitate the epigenetic programming of various tissues and organs [20]. For this reason, they are considered important signaling molecules (signalosomes) between mother and child [20,32]. Since exosomes are also able to cross the blood–brain barrier, it is possible that the positive impact of breast milk on neurodevelopment is associated with miRNAs’ activity [33].

MiRNAs are small, single-stranded, non-coding RNA molecules containing 18 to 25 nucleotides. MiRNAs are also found in plants, animals and viruses, among others [34], and they are capable of controlling up to 60% of gene expression [35,36] by inhibiting mRNA translation into protein. These particles are, thus, involved in post-transcriptional gene regulation [37,38,39]. Breast milk has been categorized as one of the biological fluids that possesses a high concentration of miRNAs encapsulated in exosomes or as free molecules, with more than 1400 distinct miRNAs identified [36,40]. Not only is human milk highly enriched in miRNAs, but it has also the highest concentration of miRNAs compared to other body fluids, including plasma [20,40]. While previous research was focused on analyzing miRNAs in the skim fraction of breast milk, recent studies investigating the lipid and cell fractions of milk have revealed a larger quantity and diversity of miRNAs compared to the skim fraction [36]. A systematic review of 30 studies on non-coding RNAs of human breast milk showed that 10 miRNAs, including miR-148a-3p, miR-30a-5p, miR-30d-5p, miR-22-3p, miR-146b-5p, miR-200a-3p, miR-200c-3p, let-7a-5p, let-7b-5p and let-7f-5p, were the most abundant miRNAs in all breast milk fractions examined [19].

Several factors have been identified to affect the miRNAs’ composition of breast milk. For example, there is evidence that the miRNAs’ concentration in human milk is influenced by the stage of lactation. Hatmall et al. reported that the total concentration of miRNA in colostrum was significantly higher than that of mature milk [40]. Similarly, Xi et al. found that the concentrations of let-7a and miRNA-378 were higher, whereas the concentration of miRNA-30B was lower, in colostrum than in mature milk [41]. On the contrary, in another study, similar levels of let-7a, mi-R16, miR-21, miR-146b, miR-181a, miR-150, and miR-223 were found across various lactation stages [42]. Differences have also been observed in species and in the expression of several miRNAs between fore- and hind-milk [18]. Although the majority of known miRNAs were identified in both pre- and post-feed milk, a number of miRNAs were found to be specific to pre-feed (*n* = 159) or post-feed milk (*n* = 180); none of the pre-feed milk-specific miRNAs was found in any post-feed samples, and vice versa [43]. Interestingly, freezer storage at −80 °C did not affect the variations in miRNAs of breast milk, indicating their stability [18].

In addition, maternal conditions such as diabetes mellitus, overweight or obesity, diet or even psychosocial factors and stress can influence the human milk miRNAs. In a study conducted by Shat et al., the levels of miRNA-148a, miRNA-30b, miRNA-let-7a, and miRNA-let-7d were found to be lower in the milk of mothers with gestational diabetes mellitus [44]. Aberrant levels of several miRNAs have also been detected in breast milk of mothers with type 1 diabetes [45]. Furthermore, in a study from Kupso et al., the expression of the majority (374 out of the 419) of miRNAs analyzed in human milk extracellular vesicles was found to correlate negatively with maternal BMI [46]. Similarly, Shah et al. showed that the exosomal content of breast milk in selected miRNAs, such as miR-148a and miR-30b, was lower by 30% and 42%, respectively, in overweight/obese mothers in comparison with a normal-weight control group [47]. In another study, 19 miRNAs including miR-575, miR-630, miR-642a-3p, and miR-652-5p, which are associated both per se and by their target genes with neurological diseases and psychological disorders, were differentially expressed in breast milk exosomes of obese nursing mothers [48]. Concerning maternal diet, animals fed an obesogenic dietary pattern exhibited higher concentrations of miR-222 and lower levels of miR-200 and miR-26 compared to the control group [49]. In humans, the expression of novel miRNAs, specifically miR-67 and miR-27, was increased in milk fat globules of women following a high-fat diet compared to those following a high-carbohydrate diet with similar calorie and protein content [50]. Additionally, maternal lifetime stress and negative life events during pregnancy were associated with the detection and expression of certain miRNAs in breast milk, such as hsa-miR-96-5p and hsa-miR-155-5p, which may be related to stress, postnatal development, and cognitive function of the offspring [51].

Concerning the association between breast milk miRNAs and improved infants’ neurodevelopment, there is evidence that several miRNAs of breast milk, such as let-7a, miR-15b miR-21, miR-29b, miR-30, miR-132, miR-138, miR-148, miR-210 and miR-574, among others, may play important roles in brain development and function [52]. For example, let-7 miRNAs are highly expressed in the developing mammalian brain and regulate neural cell proliferation and differentiation [53,54,55]. Furthermore, Walgrave et al. have demonstrated that introducing miR-132, an miRNA existing also in human milk [52], into the hippocampus of adult mice with Alzheimer’s disease restores adult hippocampal neurogenesis and improves memory deficits associated with the disease [56]. These findings highlight the potential therapeutic value of targeting miR-132 in addressing neurodegeneration. Similarly, studies in animals and humans have shown that miR-148a, which is one of the most abundant miRNAs in human milk exosomes [19], is involved in many cellular pathways, regulates neural development and exerts neuroprotective effects [18,57]. Moreover, researchers have found that several miRNAs are linked to autism spectrum disorder and may be used as potential biomarkers both for the diagnosis and prognosis of this disorder [58]. The neuroprotective effect of selected exosomal miRNAs, such as miR-21, miR-29b, miR-30 and miR-138, which can also be present in human milk, was also discussed by Nasirishargh et al. in a review article [59]. The neuroprotective effects of these miRNAs are exerted by promoting neurogenesis, neurite remodeling and survival, and neuroplasticity [59]. Especially regarding neuroplasticity, many miRNAs are involved in synaptic plasticity [60], whereas several human milk exosomal miRNAs are implicated in the gene regulation of brain synapses and in synaptic vesicle trafficking [61]. It is worth mentioning that almost half of miRNAs with possible effects on synaptic development in mammals were found to be present in the top 288 miRNAs identified in human milk exosomes [62].

Although there are studies showing that the majority of miRNAs expressed in term milk are also present in preterm milk (derived from mothers with term and preterm birth, respectively) [63,64], differences in several miRNAs have been reported between preterm and term breast milk [63,65]. For example, Shiff et al. noted higher levels of miR-148 and lower levels of miRNA-320 in both the skim and lipid fractions of colostrum samples of preterm compared to term breast milk [65]. There is also evidence that the expression patterns of nine miRNAs (miR378a-3p, miR378c, miR-378g, miR-1260a, miR-1260b, miR-4783-5p, miR-4784, miR-5787, and miR-7975) in lipid and skim fraction of preterm breast milk exhibited variations compared to those in term breast milk. The targeted genes of these miRNAs were functionally associated with elemental metabolism and lipid biosynthesis [63]. In the same study, it was also demonstrated that a total of 113 miRNAs exhibited significant differences in expression between the lipid samples of term and preterm breast milk. Among those, 68 miRNAs showed downregulation in the preterm breast milk lipid fractions, while 45 miRNAs displayed upregulation [63]. Studies in animals also provided evidence that the exosomal content of human preterm breast milk has the potential to enable tissue healing in preterms with intestinal inflammation and protect against necrotizing enterocolitis [66]. Recent evidence has also shown significant differences between preterm and term human breast milk exosomes in several miRNAs associated with brain development and neurodevelopment including miR-3196, miR-1249-3p, miR-7847-3p, miR-1908-3p and miR-23b-3p, among others [61]. Further research in this area is needed.

Overall, these findings show that miRNAs, which are well-established epigenetic modulators, are abundant in human breast milk and they are influenced by several factors relevant to lactation per se, maternal health and disease and preterm birth. They can reach the brain by crossing the blood–brain barrier, whereas several of them possess neuroprotective effects and can regulate the expression of genes implicated in infants’ brain development and function (Figure 1).

## 4. Long Non-Coding RNAs

In addition to miRNAs, breast milk also contains other types of regulatory non-coding RNAs, such as long non-coding RNAs (lncRNAs). Long non-coding RNAs are RNA molecules that are typically composed of at least 200 nucleotides [67]. They are often formed through the splicing of two or more exons derived from genomic regions located near protein-coding genes [19].

LncRNAs have a crucial role in processes such as neurogenesis, synaptogenesis, and the development of the brain (Figure 1). The utilization of high-throughput technologies has revealed their specific expression in distinct cell types, subcellular compartments, and various brain regions [68,69]. Numerous lncRNAs exhibit expression patterns that vary with age [70] and actively contribute to the determination of neural cell fate [71]. Given their involvement in these essential processes, any abnormal expression of these transcripts has the potential to lead to neurodevelopmental or neuropsychiatric disorders, including, but not limited to, autism spectrum disorder and schizophrenia [71,72].

Karlsson et al. detected 55 lncRNAs (out of 87 screened) in human milk exosomes; of them, 5 lncRNAs (CRNDE, DANCR, GAS5, SRA1 and ZFAS1) were found to be present in more than 90% of milk samples. Many of the detected lncRNAs are known to have important epigenetic roles in immune function and metabolism and are potentially related to children’s development and health [73].

Additionally, Mourtzi et al. [74] screened 88 lncRNAs in breast milk exosomes and showed that 13 lncRNAs were detected in more than 85% of milk samples, whereas 31 lncRNAs were detected in more than 50% of samples. In the same study, the expression of lncRNAs was compared between preterm and term breast milk. Differential expression analysis demonstrated at least two-fold differences in the expression of lncRNAs between the two groups, with levels of lncRNAs being higher in term breast milk as compared to the preterm one [74]. Interestingly, although the non-coding RNA activated at DNA damage (NORAD) was abundant in exosomes in both preterm and term breast milk, its expression was found to be significantly downregulated in the preterm milk exosomes. In previous studies, NORAD is a lncRNA involved in the DNA damage response and repair pathway and it is referred to as “the guardian of the human genome”. NORAD has been proven to demonstrate a protective role in mitigating brain damage, cellular apoptosis, oxidative stress, and inflammation induced by cerebral ischemia/reperfusion injury [75]. Its protective mechanism involves the regulation of miR-30a-5p and subsequent upregulation of YWHAG expression [75]. It could be suggested that utilizing lncRNAs, especially NORAD, isolated from human milk may offer a potential way to safeguard premature infants and improve their neurodevelopmental outcomes.

Another lncRNA exhibiting specific expression patterns during brain development and progenitor cell differentiation is the Sox2OT (Sox2 overlapping transcript). It has been demonstrated that by suppressing the expression of Sox2OT in mice, sepsis-induced deficits in hippocampal neurogenesis and cognitive function were improved. This improvement was achieved through the downregulation of the transcription factor SOX2. Thus, inhibiting the signaling pathway involving Sox2OT and SOX2 may hold promise as a potential therapeutic approach for treating or preventing neurological damage associated with sepsis-induced encephalopathy [76]. However, it has not yet been studied whether Sox2OT or factors inhibiting this lncRNA is/are present in human breast milk.

From the above, it is evident that, compared to miRNAs, lncRNAs of breast milk have been much less studied to date and only from an immunological and metabolic point of view. NORAD, referred to as “the guardian of the human genome” and shown to have a neuroprotective epigenetic role, was found to be abundant in human breast milk; however, it was downregulated in preterm compared to term human milk. Further studies are needed to investigate human milk non-coding RNAs related to brain development and neurodevelopment in full-term and preterm babies.

## 5. Stem Cells

Stem cells possess a remarkable capacity for both self-renewal, sustaining their undifferentiated state, and differentiation into various cell types and tissues in specific conditions [77,78,79]. In contrast, adult cells traditionally maintain their lineage commitment, yet recent studies have revealed promising approaches to induce cellular plasticity, allowing them to potentially transform into diverse cell types. This breakthrough holds significant implications for cell-based therapies in the field of regenerative medicine [79].

The discovery of stem cells within human milk dates back to 2007 [80], highlighting their presence in this unique fluid. Breastfeeding has long been recognized for its protective effects against diseases that may arise later in life, although their precise mechanism remains elusive. The presence of stem cells in both preterm and term human breast milk [81] offers one potential explanation for these beneficial effects. Interestingly, in animal studies, breast milk stem cells survive digestion and enter into the circulation and the brain, where they can be differentiated into neuronal and glial cells [82].

Stem cells from human milk contain both genetic material and bioactive molecules, such as microRNAs, which can act as epigenetic regulators [83]. The beneficial effects of breast milk stem cells may also be mediated through the paracrine action of exosomes released by these cells [84,85]. Moreover, by using the marker nestin, Cregan et al. identified nestin-positive putative stem cells in human breast milk [80]. Nestin (acronym for neuroepithelial stem cell protein) is a marker for multipotent stem cells that can differentiate into neural cells [86]. Indeed, Hosseini et al. [87] showed that human breast milk derived stem cells can differentiate into neural lineages (oligodendrocytes, astrocytes, and neurons). This differentiation capacity of milk stem cells offers valuable insights into the beneficial effects of human milk on neurodevelopment. That discovery also indicates the potential use of these cells as a suitable and easy source for cell replacement therapies targeting brain diseases. Thus, breast milk stem cells, either through their differentiation into neural cells or/and by acting as epigenetic regulators in the brain (Figure 2), seem to have opened up new horizons in the explanation of the positive short- and long-term impact of human milk. However, further research is required to elucidate their exact mechanism(s) of action after breastfeeding and define the extent of their capabilities.

## 6. Microbiome

The microbiome, which encompasses the genomes of all microorganisms, symbiotic and pathogenic, in a specific environment, has been extensively studied [88,89]. Previous assumptions regarding the existence of bacteria in human milk attributed their presence to contamination or mastitis [90,91]. However, during the early 2000s, research emerged revealing the existence of commensal bacteria in human milk and provided evidence that the DNA of these bacteria differed from that found on the surface of the breast skin, indicating that they were distinct entities [92,93,94]. By using next-generation sequencing techniques, it was found that half of the microorganism population was the same in all milk samples composing the core bacterial microbiota (bacteriome) [95]. The predominant phyla reported in human milk are Proteobacteria, Firmicutes, Actinobacteria, and Bacteroidetes. When examining the genus level, the most abundant taxa include Bifidobacterium, Lactobacillus, Streptococcus, Staphylococcus, Ralstonia, Bacteroides, Enterobacter, and Enterococcus, among others [96,97].

The composition of breast milk microbiota may be influenced by various factors. Among them, the impact of the stage of lactation on the composition of microbiota in breast milk has been investigated in several studies [98,99,100,101,102]. Findings have been inconsistent, with some studies reporting higher total bacterial loads in colostrum compared to mature milk [98,99], while others have observed an increase in bacterial loads throughout the lactation period [100,101]. On the contrary, certain studies did not detect significant alterations in bacterial numbers in breast milk samples collected within the first month after delivery, suggesting stability in microbial composition during this early period [102]. These varying results highlight the complexity and diversity of microbiota present in breast milk.

The complexity and diversity of breast milk microbiota have implications for understanding the influence of other factors on its composition. Probiotic administration during pregnancy did not influence the composition of the microbiota of breast milk, according to three separate studies involving participant sizes of 84, 125, and 20 women [97,103,104,105]. Similarly, the impact of smoking on the diversity and composition of the breast milk microbiota was examined in a study involving 393 participants, revealing no significant effects [96]. When considering milk expression methods, it was observed that using a breast pump rather than manual expression was associated with lower bacterial richness in breast milk; this could be attributed to the non-aseptic protocol used for milk collection [96].

Maternal factors, such as body mass index (BMI) and health conditions, like allergies and celiac disease, can also influence the composition of the human milk microbiota. For instance, women with higher BMI tend to exhibit a less diverse bacterial community in the breast milk microbiota, along with higher total bacterial loads and increased abundance of Lactobacillus in colostrum [106]. Nonetheless, it is important to note that in other studies, no significant impact of BMI on the composition of the breast milk microbiota was observed [96,107]. Moreover, it has been demonstrated that allergic mothers have significantly lower counts of Bifidobacteria in their breast milk compared to non-allergic mothers, as assessed using specific primers [108]. Similarly, women with celiac disease were found to have lower relative levels of Bifidobacterium and Bacteroides in their breast milk [109].

A recent cross-sectional study was carried out to examine the correlation between the milk microbiome and neurodevelopment, specifically focusing on head circumference-for-age z-scores (HCAZ) in breastfed infants [110]. Significant differences in the milk microbiota composition were found between infants with HCAZ ≥ −1 SD and HCAZ < −1 SD at both early (≤46 days postpartum) and late stages of lactation (109–184 days postpartum). The HCAZ ≥ −1 SD group had a higher abundance of Streptococcus species associated with human milk, while the HCAZ < −1 SD group, particularly at the late stage of lactation, exhibited a higher abundance of differentially abundant taxa associated with environmentally and potentially opportunistic species. These findings suggest a potential association between the milk microbiome and brain growth in breastfed infants during lactation, necessitating, however, further investigation into the interplay between the human milk microbiome and infant neurodevelopment [110].

The microbiome of breast milk shares common characteristics with the gut microbiome. After the establishment of bacterial colonization in infants, the composition of intestinal microbes becomes distinct and individualized [111]. While there is variability among individuals, the majority of these microbes can be categorized into the following four main phyla: Firmicutes, Bacteroidetes, Actinobacteria, and Proteobacteria [111]. This pattern of phyla is also observed in the microbial composition of human milk [96,97]. A study conducted by Pannaraj et al. showed that the bacterial communities of maternal milk contributed to the establishment and development of the infant gut microbiome [112]. These results emphasize the significance of the microbiome of breast milk in shaping the intestinal microbiome, including colonization with beneficial bacteria. Similarly, in the study of Solis et al., certain strains of bifidobacteria, which displayed identical genetic profiles as determined via Random Amplified Polymorphic DNA analyses, were found both in the breast milk samples of mothers and in the fecal samples of their infants taken at several time points during the first 3 months after birth. This finding suggests that there is a vertical transfer of specific bifidobacterial strains from the mother’s milk to the infant [99].

There is evidence that the gut microbiome during early life contributes to the establishment of epigenetic modifications and it is also associated with brain development and neurodevelopment [113,114,115]. The colonization of the infant’s intestine after birth, influenced by maternal flora, delivery method, early skin-to-skin contact, and neonatal diet, results in specific epigenetic patterns that can influence the protective function of the gut mucosa against future insults [116]. Furthermore, the gut microorganisms secrete molecules which can reach the brain via the circulatory system after absorption and affect the brain’s development (Figure 3), especially during sensitive periods (gut–brain axis) [117]. Interestingly, in a recent study in a humanized mouse model, the aberrant gut microbiome of preterm infants had negative effects on brain organization and maturation, and brain metabolism, as well as on behavior and memory [114]. The connection between the gut microbiome and brain function has led to investigations into its potential role in neurobehavioral disorders, such as autism spectrum disorder (ASD), anxiety and attention-deficit-hyperactivity disorder [118]. It has been reported that children with ASD have a dysbiotic microbiome with an abundance of Bacteroidetes in feces [119]. The presence of these bacteria in fecal samples could potentially explain the occurrence of gastrointestinal symptoms in certain individuals with ASD [120,121]. As neurodevelopmental impairments are often linked to the degree of prematurity, optimizing the microbial environment in early life becomes crucial for promoting healthy neurodevelopment in this vulnerable population [122]. Considering that the maternal breast milk microbiome colonizes the infant’s gut and presents similar species to the gut microbiome of the infants, it can possibly be extrapolated that mother’s breast milk microbiome also has epigenetic influences and it is associated with infants’ brain function and neurodevelopment. The precise mechanisms through which the breast milk microbiome carries out such effects on infants’ brains remain to be elucidated.

## 7. Conclusions

This article discusses the impact of breast milk bioactive factors, such as microRNAs (miRNAs), long non-coding RNAs (lncRNAs), stem cells, and the microbiome, on the neurodevelopment of preterm-born children through epigenetic mechanisms.

MiRNAs, small RNA molecules that regulate gene expression, are abundant in human milk and can be transferred to peripheral tissues through exosomes, which are resistant to digestion and capable of crossing the blood–brain barrier. These miRNAs, including let-7a, miR-15b, miR-21, miR-29b, miR-30, miR-132, miR-138, miR-148, miR-210, and miR-574, among others, play crucial roles in neurodevelopment. Additionally, human milk contains lncRNAs, such as NORAD, which exhibit protective properties against brain damage, oxidative stress, and inflammation. Stem cells, including neural stem cells, have also been identified in breast milk, contributing to its beneficial effects on neurodevelopment. Furthermore, the breast milk microbiome, composed of bacteria that seed and colonize the infant’s gut, likely shares with the gut microbiome similar effects on infant’s epigenetics and neurodevelopment. The comprehensive insights shared in this review aim to provide clarity on the link between breastfeeding and the fundamental mechanisms driving the Developmental Origins of Health and Disease (DOHaD) concept. By discussing the potential contributions of bioactive elements in breast milk, such as miRNAs, lncRNAs, stem cells, and the microbiome, to neurodevelopment through epigenetic processes, this study provides a compelling link between early life experiences and enduring health consequences in both preterm- and term-born infants.

## Figures and Tables

**Figure 1 nutrients-15-03614-f001:**
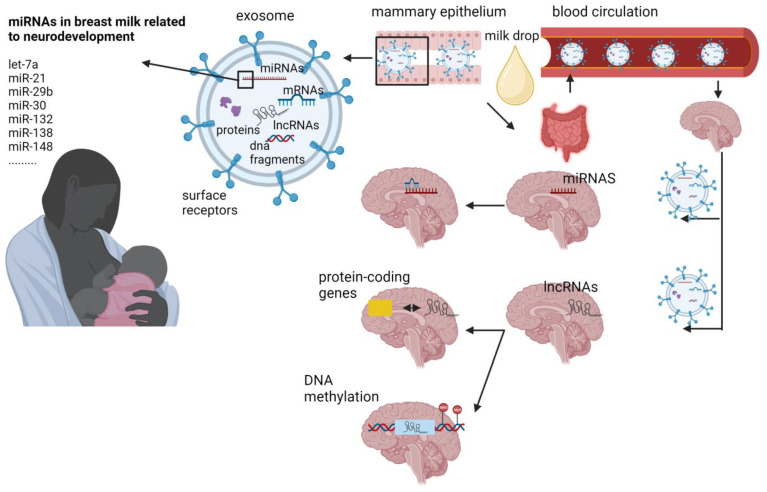
Potential mechanisms through which breast milk miRNAs and lncRNAs may be implicated in brain signaling cascade of breastfed infants. Mammary gland cells produce and release exosomes into the breast milk. Exosomes are taken up by the infant’s intestinal cells and are capable to cross the blood–brain barrier. Once inside brain cells, exosomes release their cargo (including miRNAs and lncRNAs). MiRNAs target mRNA and this binding results in modulation of gene expression. LncRNAs can interact with near protein coding genes and this interaction may involve cis-regulation of nearby genes or trans-regulation of genes in distant regions. Illustration created with BioRender.com accessed on 16 August 2023.

**Figure 2 nutrients-15-03614-f002:**
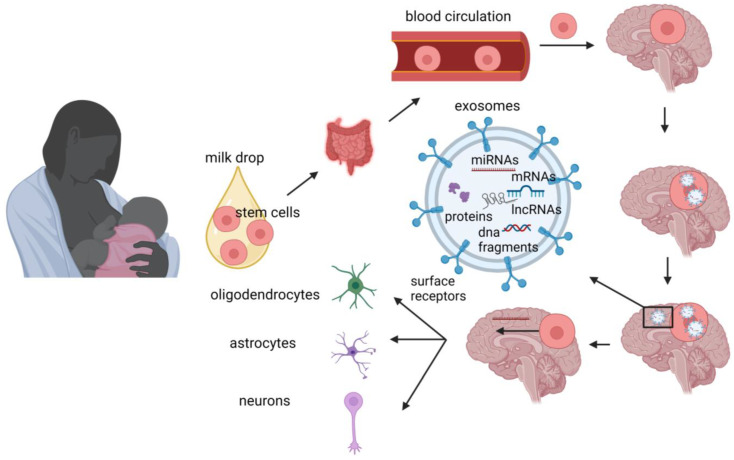
Potential mechanisms through which breast milk stem cells may exert effects on brain signaling cascade of breastfed infants. During breastfeeding, the infant ingests breast milk containing stem cells, which may cross the blood–brain barrier. Once inside the brain, stem cells may release bioactive molecules, such miRNAs, exerting epigenetic effects, and also differentiate into neural lineages. Illustration created with BioRender.com accessed on 16 August 2023.

**Figure 3 nutrients-15-03614-f003:**
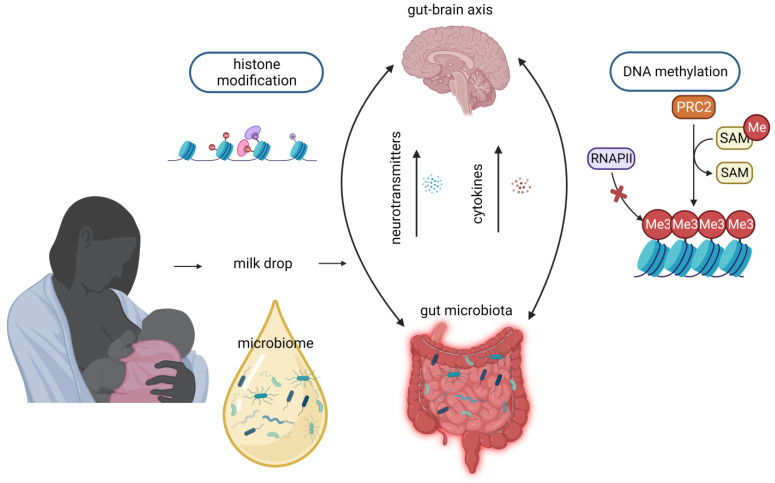
Potential mechanisms through which breast milk microbiota may exert effects on brain signaling cascades of breastfed infants. Breast milk microbiome colonizes the infant’s gut and possibly shares similar epigenetic influences on the infant’s brain. Illustration created with BioRender.com accessed on 16 August 2023.

**Table 1 nutrients-15-03614-t001:** Characteristics of the included studies.

Sources	Type of Articles	No. of Articles	Human Studies	Preclinical Studies (Mice, Pups, Rats, In Vitro, In Situ)
PubMEd Medline ScienceDirect GoogleScholar	Research articles	64	52	12
	Reviews	58		
	Total	122		

## Data Availability

Not applicable.

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
