# Peer review of "Potential Epigenetic Effects of Human Milk on Infants’ Neurodevelopment"

_nutrients, 2023, doi:10.3390/nu15163614_

Round 1
Reviewer 1 Report
The topic of this study is very interesting, as it focuses on the epigenetic effects of human milk in infants, but I still have some important questions before its acceptance for publication.
1. Please review the punctuation after citing an author and his/her collaborators. Example in line 61 lacks a period after et al.
2. In my opinion, the MS needs an illustrative outline to facilitate understanding of the potential epigenetic effects of human milk on infant neurodevelopment. The authors could present it well in the Introduction.
3. My major concern is that the article lacks material and method. They need to include at least one paragraph explaining the methodology used in this review, mentioning that it considers studies in both human and animal models, the databases used in the search for information, as well as the keywords.
Author Response
Response to Reviewer #1 Comments
The topic of this study is very interesting, as it focuses on the epigenetic effects of human milk in infants, but I still have some important questions before its acceptance for publication.
Point 1. Please review the punctuation after citing an author and his/her collaborators. Example in line 61 lacks a period after et al.
Response 1: Thank you very much for your comment. The punctuation after citing an author and his/her collaborators has been reviewed in line 61 and throughout the manuscript.
- In my opinion, the MS needs an illustrative outline to facilitate understanding of the potential epigenetic effects of human milk on infant neurodevelopment. The authors could present it well in the Introduction.
Response 2: Thank you for your comment and suggestion. In the revised manuscript, Figures have been added to facilitate understanding on the potential epigenetic effects of human milk on infants’ neurodevelopment.
- My major concern is that the article lacks material and method. They need to include at least one paragraph explaining the methodology used in this review, mentioning that it considers studies in both human and animal models, the databases used in the search for information, as well as the keywords.
Response 3: Thank you for your comment. In the revised manuscript, the Methodology of this narrative review has been added according to your suggestion.

Reviewer 2 Report
Gialeli et al. article entitled "Potential epigenetic effects of human milk on infants’ neurodevelopment" is an interesting article. In this review article, the authors provide an updated analysis of the association between breast milk feeding and infants’ neurodevelopment. The authors highlighted that such neurodevelopmental process is mediated via epigenetics process, suggesting that the components present in the breast milk could be the source neurodevelopment outcomes resulting in vulnerabilities in full-term infants and the preterm babies.
However, there are several weaknesses in the articles.
A. Abstract section:
Abstract section needs to expand bringing some molecular mechanisms in detail. Please explain what is unknown in the field. Specifically, how milk's bioactive component plays role for neurodevelopment should be highlighted. For example, how these bioactive components cross Blood Brain Barrier, which downstream signaling are impacted resulting in changes in the epigenetic signature may be included.
B. Introduction section:
Some of the historical perspective and hypothesis presented are presented well.
However, as mentioned in the abstract section, the authors should follow more detail literature search and try to explain it visually in the form of figures.
The author's own interpretation in lacking in most of the paragraph, they simply keep continuous citation of the other's work. It would be better to synthesize the view from your own perspective based on the literature analysis (the positive aspect, the controversy, negative aspect, caveats of the study).
Line 61: "Xu et al" make sure whether "et al" need to replace by "et al."
Line 150-151:"in synaptic plasticity [59] it is" - please make sure that sentence is written in grammatically correct way here and throughout the manuscript.
The authors need to provide the appropriate figures to explain the mechanism how breast milk exerts effect on the brain signaling cascade. Figures are more powerful than the words (text). I encourage to at least provide 4 figures to recapitulate and text (at least 1 figure in each subheading like miRNAs, long-coding RNAs, Stem cells, microbiome). You may also consider constructing the figures using different molecular biology tools show pathway analysis, ingenuity pathway, heatmap analysis etc.
In summary, by conceptualizing the text in figures the quality of the manuscript could immensely benefit. At this stage, I could not endorse the article without addressing the above-mentioned issues.
Author Response
Response to Reviewer #1 Comments
Gialeli et al. article entitled "Potential epigenetic effects of human milk on infants’ neurodevelopment" is an interesting article. In this review article, the authors provide an updated analysis of the association between breast milk feeding and infants’ neurodevelopment. The authors highlighted that such neurodevelopmental process is mediated via epigenetics process, suggesting that the components present in the breast milk could be the source neurodevelopment outcomes resulting in vulnerabilities in full-term infants and the preterm babies.
However, there are several weaknesses in the articles.
Point A. Abstract section:
Abstract section needs to expand bringing some molecular mechanisms in detail. Please explain what is unknown in the field. Specifically, how milk's bioactive component plays role for neurodevelopment should be highlighted. For example, how these bioactive components cross Blood Brain Barrier, which downstream signaling are impacted resulting in changes in the epigenetic signature may be included.
Response: Thank you for your comments. The abstract has been revised according to your suggestions.
- Introduction section:
Some of the historical perspective and hypothesis presented are presented well.
However, as mentioned in the abstract section, the authors should follow more detail literature search and try to explain it visually in the form of figures.
Response: Thank you for your comment and suggestion. In the revised manuscript, Figures have been added to facilitate understanding on the potential epigenetic effects of human milk on infants’ neurodevelopment.
The author's own interpretation in lacking in most of the paragraph, they simply keep continuous citation of the other's work. It would be better to synthesize the view from your own perspective based on the literature analysis (the positive aspect, the controversy, negative aspect, caveats of the study).
Response: Thank you for your comment. In the revised manuscript, we tried to provide a short interpretation after each section for miRNAs, long non-coding RNAs, stem cells and microbiome.
Line 61: "Xu et al" make sure whether "et al" need to replace by "et al."
Response: Thank you very much for your comment. The punctuation after citing an author and his/her collaborators has been reviewed in line 61 and throughout the manuscript
Line 150-151:"in synaptic plasticity [59] it is" - please make sure that sentence is written in grammatically correct way here and throughout the manuscript.
Response: Thank you very much for your comment. That sentence has been revised as follows: “Especially regarding neuroplasticity, many miRNAs are involved in synaptic plasticity [61], whereas several human milk exosomal miRNAs are implicated in gene regulation of brain synapses and in synaptic vesicle trafficking [62]. It is worth mentioning that almost half of miRNAs with possible effects on synaptic development in mammals were found to be present in the top 288 miRNAs identified in human milk exosomes [63].”
The authors need to provide the appropriate figures to explain the mechanism how breast milk exerts effect on the brain signaling cascade. Figures are more powerful than the words (text). I encourage to at least provide 4 figures to recapitulate and text (at least 1 figure in each subheading like miRNAs, long-coding RNAs, Stem cells, microbiome). You may also consider constructing the figures using different molecular biology tools show pathway analysis, ingenuity pathway, heatmap analysis etc.
In summary, by conceptualizing the text in figures the quality of the manuscript could immensely benefit. At this stage, I could not endorse the article without addressing the above-mentioned issues.
Response: Thank you once more for your pertinent comment. Figures have been added and hopefully the quality of the manuscript has been much improved.

Round 2
Reviewer 1 Report
This version presents considerable improvements.
Congratulations.
Author Response
Response to Reviewer #1 Comments
This version presents considerable improvements.
Thank you very much for your comment.

Reviewer 2 Report
I want to thank you for the authors to address the issues that I have raised. The quality of the manuscript has been improved greatly.
There are some minor issues.
- Please make sure the spacing errors, grammatical, logical flow of contents are well followed.
For example: LIne 5: "National &Kapodistrian...". Please make sure whether gmail address comply with MDPI nutrients guidelines for the 3rd author. If it does not comply, then you might either need to replace with institutional affiliation or need to omit.
Please make sure that the text in the figures is consistent in fonts, formats, size etc. Make sure that text labeling is well separated with space but not overlapping to the structure itself (Masking the structure with text may be a bit issue, try to move), if you see any such issues.
- If you feel providing flow chart of the literature selection (acceptance or rejection), some bar graph, pie chart, heat map for your analysis from the literature, you are welcome to move in that directions that will further strengthen your article visually.
Author Response
Response to Reviewer #2 Comments
I want to thank you for the authors to address the issues that I have raised. The quality of the manuscript has been improved greatly.
There are some minor issues.
-Please make sure the spacing errors, grammatical, logical flow of contents are well followed.
For example: LIne 5: "National &Kapodistrian...".
Response 1: Due to a computer issue, we were unable to open the file correctly. I believe we have now corrected all the errors.
-Please make sure whether gmail address comply with MDPI nutrients guidelines for the 3rd author. If it does not comply, then you might either need to replace with institutional affiliation or need to omit.
Response 2: We have corrected the e-mail address. gtliosis@yahoo.gr
-Please make sure that the text in the figures is consistent in fonts, formats, size etc. Make sure that text labeling is well separated with space but not overlapping to the structure itself (Masking the structure with text may be a bit issue, try to move), if you see any such issues.
Response 3: Thank you very much for your suggestions, we have corrected where necessary.
- If you feel providing flow chart of the literature selection (acceptance or rejection), some bar graph, pie chart, heat map for your analysis from the literature, you are welcome to move in that directions that will further strengthen your article visually.
Response 4: Thank you very much for your suggestion. We have added a table as per the editor's recommendation.
